# LEARNING TO GENERATE GROUNDED VISUAL CAPTIONS WITHOUT LOCALIZATION SUPERVISION

## ABSTRACT

When automatically generating a sentence description for an image or video, it often remains unclear how well the generated caption is grounded, or if the model hallucinates based on priors in the dataset and/or the language model. The most common way of relating image regions with words in caption models is through an attention mechanism over the regions that are used as input to predict the next word. The model must therefore learn to predict the attentional weights without knowing the word it should localize. This is difficult to train without grounding supervision since recurrent models can propagate past information and there is no explicit signal to force the captioning model to properly ground the individual decoded words. In this work, we help the model to achieve this via a novel cyclical training regimen that forces the model to localize each word in the image *after* the sentence decoder generates it, and then reconstruct the sentence from the localized image region(s) to match the ground-truth. Our proposed framework only requires learning one extra fully-connected layer (the localizer), a layer that can be removed at test time. We show that our model significantly improves grounding accuracy without relying on grounding supervision or introducing extra computation during inference for both image and video captioning tasks.

## 1 INTRODUCTION

Vision and language tasks, such as visual captioning, combine linguistic descriptions with data from real-world scenes. Deep learning models for such tasks have achieved great success, driven in part by the development of attention mechanisms that focus on various objects in the scene while generating captions. The resulting models, however, are known to have poor grounding performance (Liu et al., 2017), leading to undesirable behaviors such as object hallucinations (Rohrbach et al., 2018), despite having high captioning accuracy. That is, they often do not correctly associate generated words with the appropriate image regions (*e.g.,* objects) in the scene, resulting in models that lack interpretability.

Several existing approaches have tried to improve the grounding of captioning models. One class of methods generate sentence *templates* with slot locations explicitly tied to specific image regions. These slots are then filled in by visual concepts identified by off-the-shelf object detectors (Lu et al., 2018). Other methods have developed specific grounding or attention modules that aim to *attend* to the correct region(s) for generating visually groundable word. Such methods, however, rely on explicit supervision for optimizing the grounding or attention modules (Liu et al., 2017; Zhou et al., 2019) and require bounding box annotations for each visually groundable word.

In this work, we propose a novel cyclical training regimen that is able to significantly improve grounding performance without any grounding annotations. The key insight of our work is that current models use attention mechanisms conditioned on the hidden features of recurrent modules such as LSTMs, which leads to effective models with high accuracy but entangle grounding and decoding. Since LSTMs are effective at propagating information across the decoding process, the network does not necessarily need to associate particular decoded words with their corresponding image region(s). However, for a captioning model to be visually grounded, the model has to predict attentional weights without knowing the word to localize.

Based on this insight, we develop a cyclical training regimen to force the network to ground individual decoded words: *decoding → localization → reconstruction*. Specifically, the model of the decoding stage can be any state-of-the-art captioning model; in this work, we follow GVD (Zhou et al.,

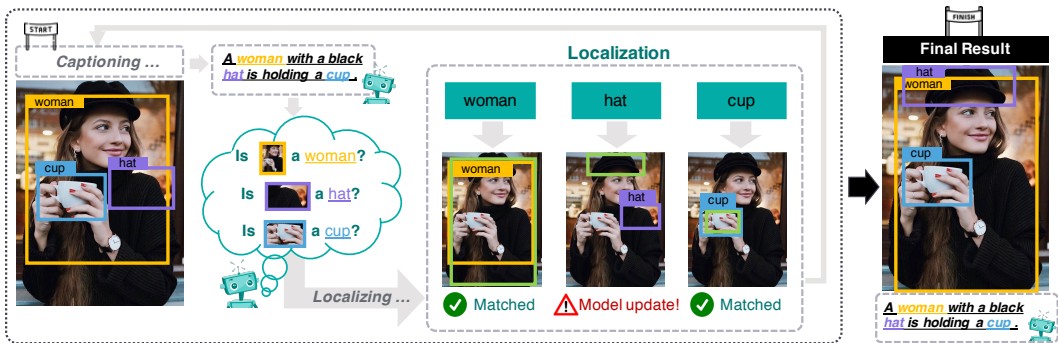

Figure 1: Visual captioning models are often not visually-grounded. As human, we perform localization to check whether the generated caption is visually-grounded. If the localized image region is incorrect, we update the model. However, without the ground-truth grounding annotation, how does the model know the localized region is incorrect? To overcome this issue, we propose to perform *localization* and *reconstruction* to regularize the captioning model to be visually-grounded without relying on the grounding annotations.

2019) to extend the widely used Up-Down model (Anderson et al., 2018). At the localization stage, each word generated by the first decoding stage is localized through a *localizer*, and the resulting grounded image region(s) are then used to reconstruct the ground-truth caption in the final stage. Both decoding and reconstruction stages are trained using a standard cross-entropy loss. Key to our method, both stages share the same decoder, thereby causing the localization stage to guide the decoder to improve its attention mechanism. Our method is simple and only adds a fully-connected layer to perform localization. During inference, we only use the (shared) decoder, thus we do not add any computational cost.

We benchmark our proposed method on the challenging Flickr30k Entities image captioning dataset (Plummer et al., 2015) and the ActivityNet-Entities video captioning dataset (Zhou et al., 2019) on both captioning and grounding performances. In addition to the existing grounding metric that calculate the grounding accuracy for each object class (Zhou et al., 2019), we further include a grounding metric that compute grounding accuracy for each generated sentence. This new metric on each sentence removes the stringency of the original evaluation metric (as we discuss in Sec. 4) and provides an alternative way of measuring the grounding performance.

Despite the simplicity of our proposed method, we are able to significantly surpass prior unsupervised models quantitatively and qualitatively on both datasets. We achieve around 18% relative improvements in terms of bridging the gap between the unsupervised baseline and supervised methods on Flickr30k Entites and around 34% on ActivityNet-Entities. We further find that our method can even outperform the supervised method on infrequent words, owing to its self-supervised nature.

**Contributions summary.** We propose object re-localization as a form of self-supervision for grounded visual captioning and present a cyclical training regimen that re-generates sentences after re-localizing the objects conditioned on each word, implicitly imposing grounding consistency. We evaluate our proposed approach on both image and video captioning tasks. We show that the proposed training regime can boost grounding accuracy over a state-of-the-art baseline, enabling grounded models to be trained without bounding box annotations, while retaining high captioning quality across two datasets and various experimental settings. Our code will be publicly released and can be found in supplemental.

## 2 RELATED WORK

**Visual captioning.** Neural models for visual captioning have received significant attention recently (Anderson et al., 2018; Ma et al., 2018; Lu et al., 2018; Donahue et al., 2015; Venugopalan et al., 2015; Rohrbach et al., 2017b; Venugopalan et al., 2017; Rohrbach et al., 2017a; Shetty et al., 2017; Park et al., 2019). Most current state-of-the-art models contain attention mechanisms, allowing the process to focus on subsets of the image when generating the next word. These attention mechanisms

can be defined over spatial locations (Vinyals et al., 2015), semantic metadata (Li et al., 2018; Yao et al., 2017; You et al., 2016; Zhou et al., 2017) or a predefined set of regions extracted via a region proposal network (Ma et al., 2018; Zanfir et al., 2016; Anderson et al., 2018; Lu et al., 2018; Das et al., 2013; Kulkarni et al., 2013). In the latter case, off-the-shelf object detectors are first used to extract object proposals (Ren et al., 2015; He et al., 2017) and the captioning model then learns to dynamically attend over them when generating the caption.

**Visual grounding.** Although attention mechanisms are generally shown to improve captioning quality and metrics, it has also been shown that they don't really focus on the same regions as a human would (Das et al., 2017). This make models less trustworthy and interpretable, and therefore creating *grounded* image captioning models, *i.e.,* models that accurately link generated words or phrases to specific regions of the image, has recently been an active research area. A number of approaches have been proposed, *e.g.,* for grounding phrases or objects from image descriptions (Rohrbach et al., 2016; Hu et al., 2016; Xiao et al., 2017; Deng et al., 2018; Zhou et al., 2019; Zhang et al., 2019), grounding visual explanations (Hendricks et al., 2018), visual co-reference resolution for actors in video (Rohrbach et al., 2017a), or improving grounding via human supervision (Selvaraju et al., 2019). Recently, Zhou et al. (2019) presented a model with self-attention based context encoding and direct grounding supervision that achieves state-of-the-art results in both the image and video tasks. They exploit ground-truth bounding box annotations to significantly improve the visual grounding accuracy. In contrast, we focus on reinforcing the visual grounding capability of the existing captioning model via a cyclical training regimen without using bounding box annotations and present a method that can increase grounding accuracy while maintaining comparable captioning performance with state of the arts.

**Cyclical training.** Cycle consistency (Wang et al., 2013; Zhu et al., 2017; He et al., 2016; Chen & Lawrence Zitnick, 2015) has been used recently in a wide range of domains, including machine translation (He et al., 2016), unpaired image-to-image translation (Zhu et al., 2017), visual question answering (Shah et al., 2019), question answering (Tang et al., 2018), image captioning (Chen & Lawrence Zitnick, 2015), video captioning (Wang et al., 2018; Duan et al., 2018), captioning and drawing (Huang et al., 2018) as well as domain adaptation (Hosseini-Asl et al., 2019). While the cyclical training regime has been explored vastly in both vision and language domains, it has not yet been used for enforcing the *visual grounding* capability of a captioning model.

## 3 METHOD

**Notation.** For a visual captioning task, we denote the input image as $I$ (or input video as $V$) and the target sentence as $S$. Each image (or video) is represented by spatial feature map(s) extracted by a ResNet-101 model and a bag of regions obtained from Faster-RCNN (Ren et al., 2015) as $\boldsymbol{R} = [\boldsymbol{r}_1, \boldsymbol{r}_2, ..., \boldsymbol{r}_N] \in \mathbb{R}^{d \times N}$. The target sentence is represented as a sequence of one-hot vectors $\boldsymbol{y}_t^* \in \mathbb{R}^s$, where $T$ is the sentence length, $t \in 1, 2, ..., T$, and $s$ is the dictionary size.

### 3.1 BASELINE

We reimplemented the model used in GVD (Zhou et al., 2019) without self-attention for region feature encoding (Ma et al., 2018; Vaswani et al., 2017) as our baseline. It is an extension of the state-of-the-art Up-Down (Anderson et al., 2018) model with the *grounding-aware region encoding* (see Appendix A.5). Specifically, our baseline model uses two LSTM modules: Attention LSTM and Language LSTM. The Attention LSTM identifies which visual representation in the image is needed for the Language LSTM to generate the next word. It encodes the global image feature $\boldsymbol{v}_g$, previous hidden state output of the Language LSTM $\boldsymbol{h}_{t-1}^L$, and the previous word embedding $e_{t-1}$ into the hidden state $\boldsymbol{h}_t^A$.

$$\boldsymbol{h}_t^A = LSTM_{Attn}([\boldsymbol{v}_g; \boldsymbol{h}_{t-1}^L; \boldsymbol{e}_{t-1}]), \quad \boldsymbol{e}_{t-1} = \boldsymbol{W}_e \boldsymbol{y}_{t-1}, \tag{1}$$

where $[; ]$ denotes concatenation, and $\boldsymbol{W}_e$ are learned parameters. We omit the Attention LSTM input hidden and cell states to avoid notational clutter in the exposition.

The Language LSTM uses the hidden state $\boldsymbol{h}_t^A$ from the Attention LSTM to dynamically attend on the bag of regions $\boldsymbol{R}$ for obtaining visual representations of the image $\hat{\boldsymbol{r}}_t$ to generate a word $y_t$.

$$z_{t,n} = \boldsymbol{W}_{aa} tanh(\boldsymbol{W}_a \boldsymbol{h}_t^A + \boldsymbol{r}_n), \quad \boldsymbol{\alpha}_t = \text{softmax}(\boldsymbol{z}_t), \quad \hat{\boldsymbol{r}}_t = \boldsymbol{R}\boldsymbol{\alpha}, \tag{2}$$

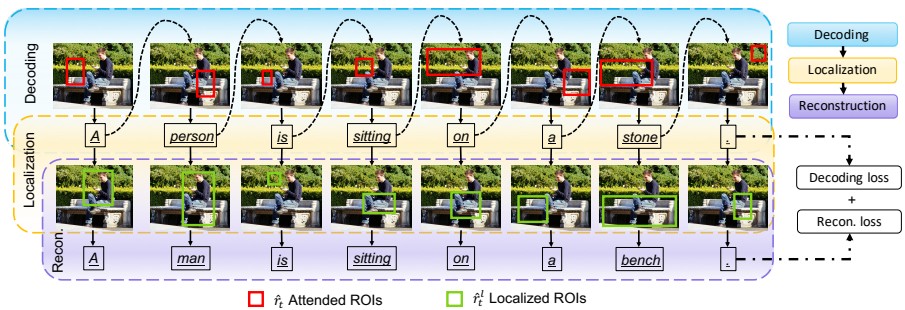

Figure 2: Proposed cyclical training regimen: *decoding → localization → reconstruction*. The decoder attends to the image regions and sequentially generate each of the output words. The localizer then uses the generated words as input to locate the image regions. Finally, the shared decoder during reconstruction stage uses the localized image regions to regenerate a sentence that matches with the ground-truth sentence.

where $\boldsymbol{W}_{aa}$ and $\boldsymbol{W}_a$ are learned parameters. The conditional probability distribution over possible output words $\boldsymbol{y}_t$ is computed as:

$$\boldsymbol{h}_t^L = LSTM_{Lang}([\hat{\boldsymbol{r}}_t, \boldsymbol{h}_t^A]), \quad p(\boldsymbol{y}_t|\boldsymbol{y}_{1:t-1}) = \text{softmax}(\boldsymbol{W}_o\boldsymbol{h}_t^L), \tag{3}$$

where $\boldsymbol{y}_{1:t-1}$ is a sequence of outputs $(\boldsymbol{y}_1, ..., \boldsymbol{y}_{t-1})$. We refer the Language LSTM and the output logit layer as the complete language decoder.

## 3.2 OVERVIEW

Our goal is to enforce the generated caption to be visually grounded, *i.e.,* attended image regions correspond specifically to individual words being generated, *without* ground-truth grounding supervision. Towards this end, we propose a novel cyclical training regimen that is comprised of *decoding*, *localization*, and *reconstruction* stages, as illustrated in Figure 2.

The intuition of our method is that the baseline network is not forced to generate a correct correspondence between the attended objects and generated words, since the LSTMs can learn priors in the data instead of looking at the image or propagate information forward which can subsequently be used to generate corresponding words in future time steps. The proposed cyclical training regimen, in contrast, aims at enforcing visual grounding to the model by requiring the language decoder (Eq. 3) to rely on the localized image regions $\hat{\boldsymbol{r}}_t^l$ to reconstruct the ground-truth sentence, where the localization is conditioned *only* on the generated word from the decoding stage. Our cyclical method can therefore be done without using any annotations of the grounding itself.

Specifically, let $\boldsymbol{y}_t^d = \mathcal{D}^d(\hat{\boldsymbol{r}}_t; \theta_d)$ be the initial language decoder with parameters $\theta_d$ (Eq. 3), trained to sequentially generate words $\boldsymbol{y}_t^d$. Let $\mathcal{G}(\boldsymbol{y}_t^d; \theta_g)$ define a *localizer* unit with parameters $\theta_g$, that learns to map (ground) each generated word to region(s) in the image, *i.e.,* $\hat{\boldsymbol{r}}_t^l = \mathcal{G}(\boldsymbol{y}_t^d, \boldsymbol{R}; \theta_g)$. Finally, let $\boldsymbol{y}_t^l = \mathcal{D}^l(\hat{\boldsymbol{r}}_t^l; \theta_l)$ be a second decoder, that is required to reconstruct the ground-truth caption using the localized region(s), instead of the attention computed by the decoder itself. We define the cycle:

$$\boldsymbol{y}_t^l = \mathcal{D}^r(\mathcal{G}(\mathcal{D}^d(\hat{\boldsymbol{r}}_t; \theta_d), \boldsymbol{R}; \theta_g); \theta_l), \quad \theta_d = \theta_l, \tag{4}$$

where $\mathcal{D}^d$ and $\mathcal{D}^l$ share parameters. Although parameters are shared, the inputs for the two language decoders differ, leading to unique LSTM hidden state values during a run. Note that the Attention LSTMs and logit layers in the two stages also share parameters, though they are omitted for clarity.

Through cyclical joint training, both $\mathcal{D}^d$ and $\mathcal{D}^l$ are required to generate the same ground-truth sentence. They are both optimized to maximize the likelihood of the correct caption:

$$\theta^* = \arg\max_{\theta_d} \sum \log p(\boldsymbol{y}_t^d; \theta_d) + \arg\max_{\theta_l} \sum \log p(\boldsymbol{y}_t^l; \theta_l), \tag{5}$$

During training, the localizer regularizes the region attention of the reconstructor and the effect is further propagated to the baseline network in the decoding stage, since the parameters of Attention

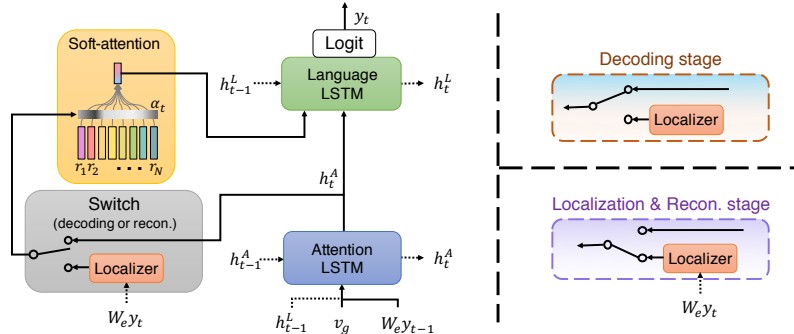

Figure 3: Proposed model architecture (left) and how the model operates during decoding, localization, and reconstruction stages (right). During the decoding stage, the soft-attention module uses the hidden state of the Attention LSTM to compute attention weights on image regions. During the localization and reconstruction stage, the soft-attention module instead uses the generated word from decoding stage to compute attention weights on image regions.

LSTM and Language LSTM are shared for both decoding and reconstruction stages. Note that the gradient from reconstruction loss will not backprop to the decoder $\mathcal{D}^d$ in the decoding stage since the generated words used as input to the localizer are leafs in the computational graph. The network is implicitly regularized to update its attention mechanism to match with the localized image regions $\hat{r}_t \mapsto \hat{r}_t^l$. In Sec. 4.3, we demonstrate that the localized image regions $\hat{r}_t^l$ indeed have higher attention accuracy than $\hat{r}_t$ when using ground-truth words as inputs for the localizer.

### 3.3 CYCLICAL TRAINING

We now describe each stage of our cyclical model in detail, as illustrated in Figure 3.

**Decoding.** We first use the baseline model presented in Sec. 3.1 to generate a sequence of words $\boldsymbol{y} = [\boldsymbol{y}_1^d, \boldsymbol{y}_2^d, ..., \boldsymbol{y}_T^d]$, where $T$ is the ground-truth sentence length.

**Localization.** Following the decoding process, a localizer $\mathcal{G}$ is then learned to localize the image regions from each generated word $\boldsymbol{y}_t$.

$$\boldsymbol{e}_t = \boldsymbol{W}_e \boldsymbol{y}_t^d, \quad z_{t,n}^l = (\boldsymbol{W}_l \boldsymbol{e}_t)^\top \boldsymbol{r}_n \quad \text{and} \quad \boldsymbol{\beta}_t = \text{softmax}(\boldsymbol{z}_t^l), \tag{6}$$

where $\boldsymbol{e}_t$ is the embedding for the word generated during decoding stage at step $t$, $\boldsymbol{r}_n$ is the image representation of a region proposal, and $\boldsymbol{W}_e$ and $\boldsymbol{W}_l$ are the learned parameters. Based on the localized weights $\boldsymbol{\beta}_t$, the localized region representation can be obtained by $\hat{\boldsymbol{r}}_t^l = \boldsymbol{R}\boldsymbol{\beta}$.

**Reconstruction.** Finally, the shared language decoder $\mathcal{D}^l$ relies on the localized region representation $\hat{\boldsymbol{r}}_t^l$ to generate the next word. The probability over possible output words is:

$$\boldsymbol{h}_t^L = LSTM_{Lang}([\hat{\boldsymbol{r}}_t^l; \boldsymbol{h}_t^A]), \quad p(\boldsymbol{y}_t^l | \boldsymbol{y}_{1:t-1}^l) = \text{softmax}(\boldsymbol{W}_o \boldsymbol{h}_t^L), \tag{7}$$

Given the target ground truth caption $\boldsymbol{y}_{1:T}^*$ and our proposed captioning model parameterized with $\theta$, we minimize the following cross-entropy losses:

$$\mathcal{L}_{CE}(\theta) = -\lambda_1 \underbrace{\sum_{t=1}^{T} log(p_\theta(\boldsymbol{y}_t^* | \boldsymbol{y}_{1:t-1}^*)) \mathbb{1}_{(\boldsymbol{y}_t^* = \boldsymbol{y}_t^d)}}_{\text{decoding loss}} -\lambda_2 \underbrace{\sum_{t=1}^{T} log(p_\theta(\boldsymbol{y}_t^* | \boldsymbol{y}_{1:t-1}^*)) \mathbb{1}_{(\boldsymbol{y}_t^* = \boldsymbol{y}_t^l)}}_{\text{reconstruction loss}} \tag{8}$$

where $\lambda_1$ and $\lambda_2$ are weighting coefficient selected on the validation split.

## 4 EXPERIMENTS

**Datasets.** We use the Flickr30k Entities image dataset (Plummer et al., 2015) and the ActivityNet-Entities video dataset (Zhou et al., 2019) for evaluating our proposed approach. Flickr30k Entities

contains 275k annotated bounding boxes from 31k images associated with natural language phrases. Each image is annotated with 5 crowdsourced captions. ActivityNet-Entities contains 15k videos with 158k spatially annotated bounding boxes from 52k video segments.

**Captioning evaluation metrics.** We measure captioning performance using four language metrics, including BLEU (Papineni et al., 2002), METEOR (Banerjee & Lavie, 2005), CIDEr (Vedantam et al., 2015), and SPICE (Anderson et al., 2016).

**Grounding evaluation metrics.** Following the grounding evaluation from GVD (Zhou et al., 2019), we measure the attention accuracy on generated sentences, denoted by $F1_{all}$ and $F1_{loc}$. In $F1_{all}$, a region prediction is considered correct if the object word[1] is correctly predicted and also correctly localized. We also compute $F1_{loc}$, which only considers correctly-predicted object words. Please see illustration of the grounding metrics in Appendix A.1.

In the original formulation, the precision and recall for the two F1 metrics are computed **for each object class**, and it is set to zero if an object class has never been predicted. The scores are computed for each object class and averaged by the total number of classes. Such metrics are extremely stringent as captioning models are generally biased toward certain words in the vocabulary, given the long-tailed distribution of words. In fact, both the baseline and proposed method generate about 45% of the annotated object words within the val set in Flickr30k Entities. The grounding accuracy of the other 55% of the classes are therefore zero, making the averaged grounding accuracy seemingly low.

**Measuring grounding per generated sentence.** Instead of evaluating grounding on each object class (which might be less intuitive), we include a new grounding evaluation metric *per sentence* to directly reflect the grounding measurement of each generated sentence. The metrics are computed against a pool of object words and their ground-truth bounding boxes (GT bbox) collected across five GT captions on Flickr30k Entities (and one GT caption on ActivityNet-Entities). We use the same $Prec_{all}$, $Rec_{all}$, $Prec_{loc}$, and $Rec_{loc}$ as defined previously, but their scores are averaged on each of the generated sentence. As a result, the $F1_{loc\_per\_sent}$ measures the F1 score only on the generated words. The model will not be punished if some object words are not generated, but it also needs to maintain diversity to achieve high captioning performance.

## 4.1 IMPLEMENTATION AND TRAINING DETAILS

**Region proposal and spatial features.** Following GVD (Zhou et al., 2019), we extracted 100 region proposals from each image (video frame) and encode them via the *grounding-aware region encoding*. Please refer to Appendix A.5 for more implementation details.

**Training.** We train the model with ADAM optimizer (Kingma & Ba, 2015). The initial learning rate is set to $1e - 4$. Learning rates automatically drop by 10x when the CIDEr score is saturated. The batch size is 32 for Flickr30k Entities and 96 for ActivityNet-Entities. We learn the word embedding layer from scratch for fair comparisons with existing work (Zhou et al., 2019).

## 4.2 CAPTIONING AND GROUNDING PERFORMANCE COMPARISON

**Flickr30k Entities.** We first compare the proposed method with our baseline with or without grounding supervision on the Flickr30k Entities test set (see Table 1). To train the supervised baseline, we train the attention mechanism as well as add the region classification task using the ground-truth grounding annotation, similar to GVD (Zhou et al., 2019). We train the proposed baselines and our method on the training set and choose the best performing checkpoints based on their CIDEr score on the val set. Our experimental results are reported by averaging across five runs on the test set. We report only the mean of the five runs to keep the table uncluttered. When compared to the existing state of the arts, our proposed baselines achieve comparable captioning evaluation performances and grounding accuracy. Using the resulting supervised baseline as the upper bound, our proposed method with cyclical training statistically achieves around 20 to 25% relative grounding accuracy improvements for both $F1_{all}$ and $F1_{loc}$ and 10 to 15% for $F1_{all\_per\_sent}$ and $F1_{loc\_per\_sent}$ without utilizing any grounding annotations or additional computation during inference.

**ActivityNet-Entities.** We adapt our proposed baselines and method to the ActivityNet-Entities video dataset (see Table 2). We can see that our proposed method significantly improved the grounding

---

[1]The object words are words in the sentences that are annotated with corresponding image regions.

| Method | Grounding supervision | Captioning Evaluation | | | | | Grounding Evaluation | | | |
|---|---|---|---|---|---|---|---|---|---|---|
| | | B@1 | B@4 | M | C | S | F1$_{all}$ | F1$_{loc}$ | F1$_{all\_per\_sent}$ | F1$_{loc\_per\_sent}$ |
| ATT-FCN | | 64.7 | 19.9 | 18.5 | - | - | - | - | - | - |
| NBT | | 69.0 | 27.1 | 21.7 | 57.5 | 15.6 | - | - | - | - |
| Up-Down | | 69.4 | 27.3 | 21.7 | 56.6 | 16.0 | 4.14 | 12.3 | - | - |
| GVD (w/o SelfAttn) | | 69.2 | 26.9 | 22.1 | 60.1 | 16.1 | 3.97 | 11.6 | - | - |
| GVD | ✓ | 69.9 | 27.3 | 22.5 | 62.3 | 16.5 | 7.77 | 22.2 | - | - |
| Baseline* | ✓ | 69.0 | 26.8 | 22.4 | 61.1 | 16.8 | 8.44 (+100%) | 22.78 (+100%) | 27.37 (+100%) | 63.19 (+100%) |
| Baseline* | | 69.1 | 26.0 | 22.1 | 59.6 | 16.3 | 4.08 (+0%) | 11.83 (+0%) | 13.20 (+0%) | 31.83 (+0%) |
| Cyclical* | | **69.4** | 26.9 | **22.3** | 60.8 | 16.6 | **5.11** (+24%) | **14.15** (+21%) | **15.15** (+14%) | **35.56** (+12%) |

Table 1: Performance comparison on the Flickr30k Entities **test set**: ATT-FCN (You et al., 2016), NBT (Lu et al., 2018), Up-Down (Anderson et al., 2018), GVD (Zhou et al., 2019), and Baseline is our reimplementation of GVD. *: our results are averaged **across five runs**. Only numbers reported by multiple runs are considered to be bolded.

| Method | Grounding supervision | Captioning Evaluation | | | | | Grounding Evaluation | | | |
|---|---|---|---|---|---|---|---|---|---|---|
| | | B@1 | B@4 | M | C | S | F1$_{all}$ | F1$_{loc}$ | F1$_{all\_per\_sent}$ | F1$_{loc\_per\_sent}$ |
| GVD | | 23.0 | 2.27 | 10.7 | 44.6 | 13.8 | 0.28 | 1.13 | - | - |
| GVD (w/o SelfAttn) | | 23.2 | 2.28 | 10.9 | 45.6 | 15.0 | 3.70 | 12.7 | - | - |
| GVD | ✓ | 23.9 | 2.59 | 11.2 | 47.5 | 15.1 | 7.11 | 24.1 | - | - |
| Baseline* | ✓ | 23.1 | 2.13 | 10.7 | 45.0 | 14.6 | 7.30 (+100%) | 25.02 (+100%) | 17.88 (+100%) | 60.23 (+100%) |
| Baseline* | | 23.2 | 2.22 | 10.8 | 45.9 | **15.1** | 3.75 (+0%) | 12.00 (+0%) | 9.41 (+0%) | 31.68 (+0%) |
| Cyclical* | | **23.7** | **2.45** | **11.1** | **46.4** | 14.8 | **4.68** (+26%) | **15.84** (+29%) | **12.60** (+38%) | **44.04** (+43%) |

Table 2: Performance comparison on the ActivityNet-Entities **val set**: GVD (Zhou et al., 2019) and Baseline is our reimplementation of GVD. *: our results are averaged **across five runs**. Only numbers reported by multiple runs are considered to be bolded.

accuracy around 25% to 30% relative grounding accuracy improvements for both $F1_{all}$ and $F1_{loc}$ and around 40% for $F1_{all\_per\_sent}$ and $F1_{loc\_per\_sent}$.

### 4.3 ANALYSIS

**Are localized image regions better than attended image regions during training?** Given our intuition described in Sec. 3, we expect the decoder to be regularized to update its attention mechanism to match with the localized image regions $\hat{r}_t \mapsto \hat{r}_t^l$. This indicates that the localized image regions should be more accurate than the attended image regions by the decoder during training. To verify this, we compute the attention accuracy for both decoder and localizer over ground-truth sentences following (Rohrbach et al., 2016; Zhou et al., 2018). The attention accuracy for localizer is 20.4% and is higher than the 19.3% from the decoder at the end of training, which confirms our hypothesis.

**Grounding performance when using a *better* object detector.** In Table 1 and 2 we showed that our proposed method significantly improved the grounding accuracy for both image and video captioning. These experimental settings follow the widely used procedure for visual captioning systems: extract regional proposal features and generate visual captions by attending to those extracted visual features.

| # | Grounding supervision | Captioning Eval. | | | Grounding Eval. | | F1$_{loc\_per\_sent}$ |
|---|---|---|---|---|---|---|---|
| | | M | C | S | F1$_{all}$ | F1$_{loc}$ | |
| **Unrealistically perfect object detector** | | | | | | | |
| Baseline | ✓ | 25.3 | 76.5 | 22.3 | 23.19 | 52.83 | 90.76 |
| Baseline | | 25.2 | 76.3 | 22.0 | 20.82 | 48.74 | 77.81 |
| Cyclical | | **25.8** | **80.2** | **22.7** | **25.27** | **54.54** | **81.56** |
| **Grounding-biased object detector** | | | | | | | |
| Baseline | ✓ | 21.3 | 53.3 | 15.5 | 8.23 | 23.95 | 66.96 |
| Baseline | | **21.2** | **52.4** | **15.4** | 5.95 | 17.51 | 42.84 |
| Cyclical | | **21.2** | 52.0 | **15.4** | **6.87** | **19.65** | **50.25** |

Table 3: Grounding performance when using *better* object detector on the Flickr30k Entities **test** set (see Table 4 for complete version).

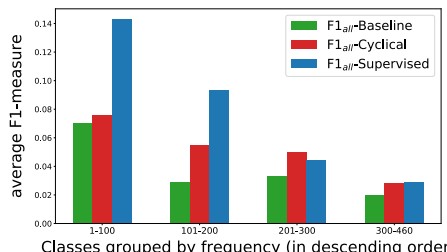

Figure 4: Average F1$_{all}$-score per class as a function of class frequency.

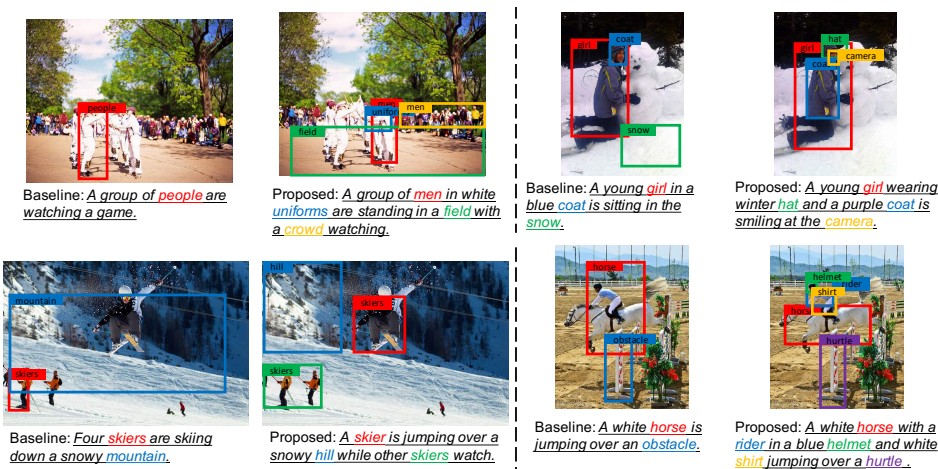

Figure 5: Generated captions and corresponding visual grounding regions with comparison between baseline (left) and proposed approach (right). Our proposed method is able to generate more descriptive sentences while selecting the correct regions for generating the corresponding words.

One might ask, what if we have a better object detector that can extract robust visual representation that are better aligned with the word embeddings? Will visual grounding still an issue for captioning?

To answer this, we ran two sets of experiments (Table 3): **(1) *Perfect* object detector**: we replace the ROIs by ground-truth bbox and represent the new ROIs by learning embedding features directly from ground-truth object words associated with each ground-truth bbox. This experiment gives an estimate of the captioning and grounding performance if we have (almost) perfect ROI representations (though unrealistic). We can see that the fully-supervised method achieves an $F1_{all}$ of only 23%, which further confirms the difficulty of the metric and the necessity of our grounding metric on a per sentence level (note that $F1_{loc\_per\_sent}$ shows 90%). We can also see that baseline (unsup.) still leaves room for improvement on grounding performance. Surprisingly, our method improved both captioning and grounding accuracy and surpasses the fully-supervised baseline except on the $F1_{loc\_per\_sent}$. We find that it is because the baseline (sup.) overfits to the training set, while ours is regularized from the cyclical training. Also, our generated object words are more diverse, which is critical for $F1_{all}$ and $F1_{loc}$. **(2) *Grounding-biased* object detector**: we extract ROI features from an object detector pre-trained on Flickr30k. Thus, the ROI features and their associated object predictions are biased toward the annotated object words but do not generalize to predict diverse captions compared to the original object detector trained from Visual Genome, resulting in lower captioning performance. We can see that our proposed method still successfully improves grounding and maintains captioning performance in this experiment setting as well.

**How does the number of annotations affect grounding performance?** In Figure 4, we present the average F1-score on the Flickr30k Entities val set when grouping classes according to their frequency of appearance in the training set[3]. We see that, unsurprisingly, the largest difference in grounding accuracy between the supervised and our proposed cyclical training is for the 50 most frequently appearing object classes, where enough training data exists. As the number of annotated boxes decreases, however, the difference in performance diminishes, and cyclical training appears to be more robust. Overall, we see that the supervised method is biased towards frequently appearing objects, while grounding performance for the proposed approach is more balanced among classes.

**Qualitative analysis.** We additionally conduct qualitative analysis for comparing the baseline (Unsup.) and the proposed method in Figure 5. Each highlighted word has a corresponding image region annotated on the original image. The image regions are selected based on the region with the maximum attention weight in $\alpha_t$. We can see that our proposed method significantly outperformed the baseline (Unsup.) in terms of both the quality of the generated sentence and grounding accuracy. In addition, we also discuss a number of correct and incorrect examples of our proposed method in Figure 8 in the Appendix. Please refer to the Appendix A.4 for further discussions on the qualitative results and the complete sequence of attended image regions of examples in Figure 5.

---

[3]We group the 460 object classes in 10 groups, sorted by the number of annotated bounding boxes.

## 5 CONCLUSION

Working from the intuition that typical attentional mechanisms in the visual captioning task are not forced to ground generated words since recurrent models can propagate past information, we devise a novel cyclical training regime to explicitly force the model to ground each word without grounding annotations. Our method only adds a fully-connected layer during training, which can be removed during inference, and we show thorough quantitative and qualitative results demonstrating around 20% or 30% relative improvements in visual grounding accuracy over existing methods for image and video captioning tasks.

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

| Method | Grounding supervision | Captioning Evaluation | | | | | Grounding Evaluation | | | |
|---|---|---|---|---|---|---|---|---|---|---|
| | | B@1 | B@4 | M | C | S | F1$_{all}$ | F1$_{loc}$ | F1$_{all\_per\_sent}$ | F1$_{loc\_per\_sent}$ |
| **Unrealistically perfect object detector** | | | | | | | | | | |
| Baseline | ✓ | 75.6 | 32.0 | 25.3 | 75.6 | 22.3 | 23.19 (+100%) | 52.83 (+100%) | 51.43 (+100%) | 90.76 (+100%) |
| Baseline | | 75.1 | 32.1 | 25.2 | 76.3 | 22.0 | 20.82 (+0%) | 48.74 (+0%) | 43.21 (+0%) | 77.81 (+0%) |
| Cyclical | | **76.7** | **32.8** | **25.8** | **80.2** | **22.7** | **25.27** (+188%) | **54.54** (+142%) | **46.98** (+46%) | **81.56** (+29%) |
| **Grounding-biased object detector** | | | | | | | | | | |
| Baseline | ✓ | 65.9 | 23.4 | 21.3 | 53.3 | 15.5 | 8.23 (+100%) | 23.95 (+100%) | 28.06 (+100%) | 66.96 (+100%) |
| Baseline | | **66.1** | **23.5** | **21.2** | **52.4** | **15.4** | 5.95 (+0%) | 17.51 (+0%) | 18.11 (+0%) | 42.84 (+0%) |
| Cyclical | | 65.5 | 23.3 | **21.2** | 52.0 | **15.4** | **6.87** (+40%) | **19.65** (+33%) | **20.82** (+27%) | **50.25** (+31%) |

Table 4: Grounding performance when using better object detector on the Flickr30k Entities **test** set (results are averaged three runs). Fully-supervised method (Sup.) is used as upper bound, thus its numbers are not bolded.

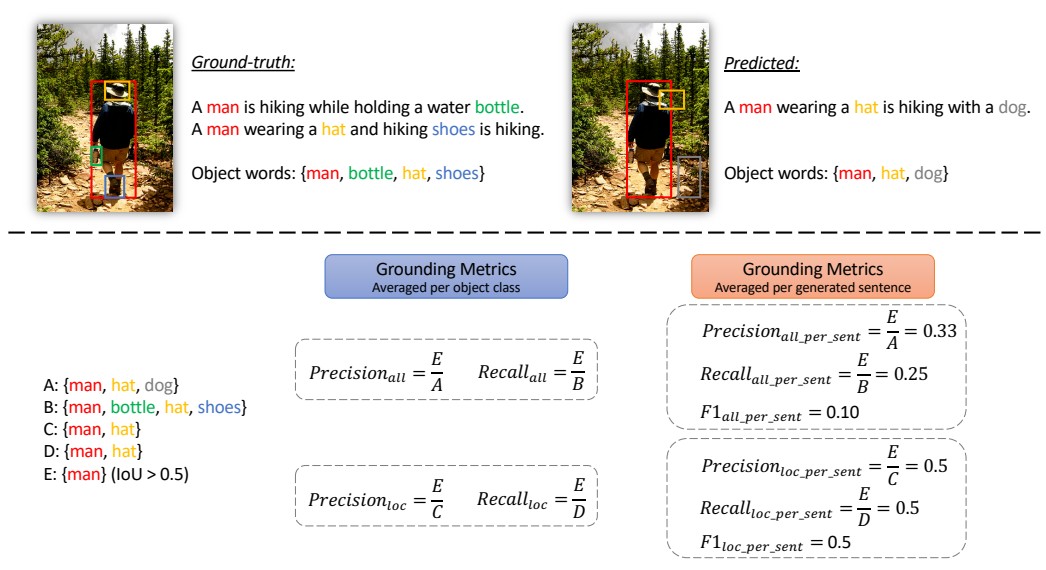

Figure 6: Illustration of Grounding metrics.

# A  APPENDIX

## A.1  GROUNDING EVALUATION METRICS ILLUSTRATED

To help better understand the grounding evaluation metrics used in this work, we illustrated the grounding evaluation metrics in Figure 6.

We define the number of object words in the generated sentences as A, the number of object words in the GT sentences as B, the number of correctly predicted object words in the generated sentences as C and the counterpart in the GT sentences as D, and the number of correctly predicted and localized words as E (see illustration of the grounding metrics in Appendix A.1). A region prediction is considered correct if the object word is correctly predicted and also correctly localized (*i.e.,* IoU with GT box > 0.5). We then compute two version of the precision and recall as Prec$_{all}$ = $\frac{E}{A}$, Rec$_{all}$ = $\frac{E}{B}$, Prec$_{loc}$ = $\frac{E}{C}$, and Rec$_{loc}$ = $\frac{E}{D}$.

The original grounding evaluation metric proposed in GVD (Zhou et al., 2019) average the grounding for each object class. We additionally calculate the grounding accuracy for each generated sentence as demonstrated in the figure. From this example, we can see that while $Precision_{all}$ counts *dog* as a wrong prediction for the *dog* object class, the $Precision_{loc}$ only cares if *man* and *hat* are predicted and correctly localizer (IoU > 0.5).

| | Captioning Eval. | | | Grounding Eval. | |
|---|---|---|---|---|---|
| # | M | C | S | F1$_{\text{all}}$ | F1$_{\text{loc}}$ |
| Baseline (Unsup.) | **22.3** | 62.1 | 16.0 | 4.18 | 11.9 |
| Cyclical | 22.2 | **62.2** | **16.2** | **5.63** | **14.6** |
| - Attention consistency | **22.3** | 61.8 | **16.2** | 4.19 | 11.3 |
| - Localizer using $\boldsymbol{h}^A$ | 22.2 | 61.8 | 16.1 | 4.58 | 11.3 |

Table 5: Model ablation study on the Flickr30k Entities val set.

| | Captioning Evaluation | | | | | Grounding Evaluation | | | |
|---|---|---|---|---|---|---|---|---|---|
| Method | B@1 | B@4 | M | C | S | F1$_{\text{all}}$ | F1$_{\text{loc}}$ | F1$_{\text{all\_per\_sent}}$ | F1$_{\text{loc\_per\_sent}}$ |
| Baseline | 69.1 | 26.0 | 22.1 | 59.6 | 16.3 | 4.08 | 11.83 | 13.20 | 31.83 |
| Cyclical | 69.4 | 26.9 | 22.3 | 60.8 | **16.6** | 5.11 | 14.15 | 15.15 | 35.56 |
| Cyclical (1) | 69.7 | 27.0 | 22.2 | 60.1 | 16.5 | **5.14** | **14.32** | 15.36 | 36.33 |
| Cyclical (2) | **69.9** | **27.5** | **22.4** | **62.0** | **16.6** | 5.13 | 13.99 | **16.30** | **38.45** |

Table 6: Performance comparison on the Flickr30k Entities **test set**. All results are averaged **across five runs**.

## A.2 ADDITIONAL ANALYSIS

**Should we explicitly make attended image regions to be similar to localized image regions?** One possible way to regularize the attention mechanism of the decoder is to explicitly optimize $\hat{r}_t \mapsto \hat{r}_t^l$ via KL divergence over two soft-attention weights $\boldsymbol{\alpha}_t$ and $\boldsymbol{\beta}_t$. The experimental results are shown in Table 5 (*Attention consistency*). We use a single run unsupervised baseline with a fix random seed as baseline model for ablation study. We can see that when explicitly forcing the attended regions to be similar to the localized regions, both the captioning performance and the grounding accuracy remain similar to the baseline (unsup.). We conjecture that this is due to the noisy localized regions at the initial training stage. When forcing the attended regions to be similar to noisy localized regions, the Language LSTM will eventually learn to not rely on the attended region at each step for generating sequence of words. To verify, we increase the weight for attention consistency loss and observed that it has lower grounding accuracy (F1$_{\text{all}}$ = 3.2), but the captioning will reach similar performance while taking 1.5x longer to reach convergence.

**Is using only the generated word for localization necessarily?** Our proposed localizer (Eq. 6 and Figure 3) relies on purely the word embedding representation to locate the image regions. This forces the localizer to rely only on the word embedding without biasing it with the memorized information from the Attention LSTM. As shown in the Table 5 (localizer using $\boldsymbol{h}^A$), although this achieves comparable captioning performance, it has lower grounding accuracy improvement compared to our proposed method.

**Can words that are not visually-groundable handled differently?** In the proposed method, all the words are handled the same regardless of whether they are visually-groundable or not. Yet, typically words that are nouns or verbs are more likely to be grounded, and words like "a", "the", *etc*, are not visually-groundable.

We explored a few method variants to handle nouns and verbs differently. Mainly, we explored with two variants. Cyclical (1): the reconstruction loss is only computed when the target word is either nouns or verbs. Cyclical (2): the localized region representation will be invalid (set to zero) if the target word is neither nouns nor verbs.

The experimental results are shown in Table 6, 7, and 8. For the first variant, Cyclical (1), we observed that the captioning performance stays the same while grounding accuracy has a small improvement. On the other hand, for the second variant, Cyclical (2), we can see that all captioning scores are improved over baseline with CIDEr improved 2.4. We can also see that grounding accuracy on per sentence basis further improved as well. We then conducted further experiments on both ActivityNet-Entities and Flickr30k Entities with *unrealistically perfect object detector*, but the improvements however are not consistent. In summary: on the Flickr30k Entities test set, we observed that CIDEr is better and grounding per sentence better, on the ActivityNet-Entities val set,

| Method | Captioning Evaluation | | | | | Grounding Evaluation | | | |
|---|---|---|---|---|---|---|---|---|---|
| | B@1 | B@4 | M | C | S | $F1_{all}$ | $F1_{loc}$ | $F1_{all\_per\_sent}$ | $F1_{loc\_per\_sent}$ |
| Baseline | 23.2 | 2.22 | 10.8 | 45.9 | **15.1** | 3.75 | 12.00 | 9.41 | 31.68 |
| Cyclical | 23.7 | 2.45 | 11.1 | 46.4 | 14.8 | **4.68** | **15.84** | **12.60** | **44.04** |
| Cyclical (2) | **23.9** | **2.58** | **11.2** | **46.6** | 14.8 | 4.48 | 15.01 | 11.53 | 40.30 |

Table 7: Performance comparison on the ActivityNet-Entities **val set**. All results are averaged **across five runs**.

| Method | Captioning Evaluation | | | | | Grounding Evaluation | | | |
|---|---|---|---|---|---|---|---|---|---|
| | B@1 | B@4 | M | C | S | $F1_{all}$ | $F1_{loc}$ | $F1_{all\_per\_sent}$ | $F1_{loc\_per\_sent}$ |
| **Unrealistically perfect object detector** | | | | | | | | | |
| Baseline | 75.1 | 32.1 | 25.2 | 76.3 | 22.0 | 20.82 | 48.74 | 43.21 | 77.81 |
| Cyclical | **76.7** | **32.8** | **25.8** | **80.2** | **22.7** | 25.27 | 54.54 | 46.98 | 81.56 |
| Cyclical (2) | 75.8 | 32.2 | 25.6 | 79.0 | 22.4 | **25.65** | **55.81** | **48.99** | **85.99** |

Table 8: Grounding performance when using better object detector on the Flickr30k Entities **test** set (results are averaged three runs).

the captioning performances are about the same but grounding accuracy became worse, and on the Flickr30k Entities test set with unrealistically perfect object detector, captioning performances are slightly worse but grounding accuracy improved.

### A.3 HUMAN EVALUATION ON GROUNDING

We conduct a human evaluation on the perceptual quality of the grounding. We asked 10 human subjects to pick the best among two grounded regions (by baseline and Cyclical) for each word. The subjects have three options to choose from: 1) grounded region A is better, 2) grounded region B is better, and 3) they are about the same (see Figure 7 for example). Authors or other colleagues familiar with the proposed method were excluded from the study. Each of the human subjects were given 25 images, each with a varying number of groundable words. Each image was presented to two different human subjects in order to be able to measure inter-rater agreement. To avoid being biased towards the object words defined in the dataset for automatic grounding evaluation, for the study we define a word to be groundable if it is either a noun or verb. The order of approaches was randomized for each sentence.

Our experiment on the Flickr30k Entities val set showed that: 28.1% of words are more grounded by Cyclical, 24.8% of words are more grounded by baseline, and 47.1% of words are similarly grounded.

We also measured inter-rater agreement between each pair of human subjects: 72.7% of ratings are the same, 4.9% of ratings are the opposite, and 22.4% of ratings could be ambiguous (*e.g.,* one chose A is better, the other chose they are about the same).

We would also like to make a note that the grounded words judged to be similar largely consisted of very easy or impossible cases. For example, words like *mountain, water, street, etc,* are typically rated to be "about the same" since they usually have many possible boxes and is very easy for both models to ground the words correctly. On the other hand, for visually ungroundable cases, *e.g., stand* appears a lot and the subject would choose *about the same* since the image does not cover the fact that the person's feet are on the ground.

We see that the human study results follow the grounding results presented in the paper and show an improvement in grounding accuracy for the proposed method over a strong baseline. The improvement is achieved without grounding annotations or extra computation at test time.

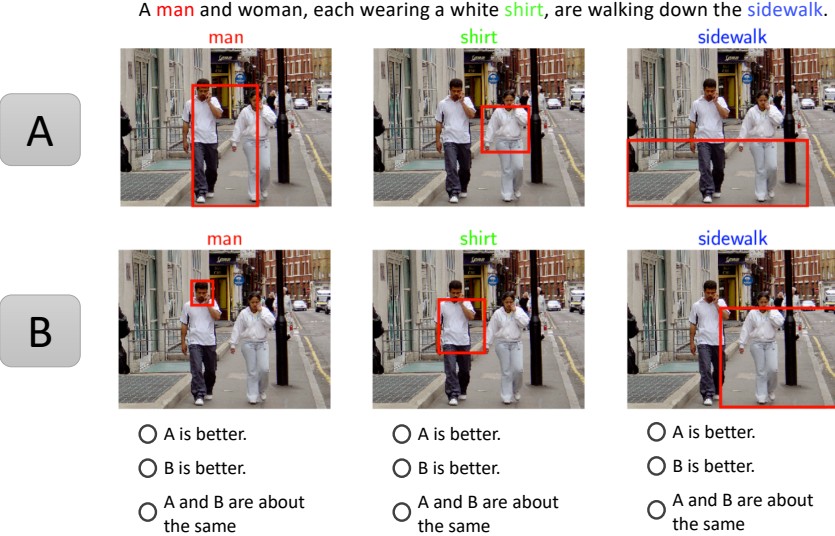

Figure 7: Demonstration of our human evaluation study on grounding. Each human subject is required to rate which method (A or B) has a better grounding on each highlighted word.

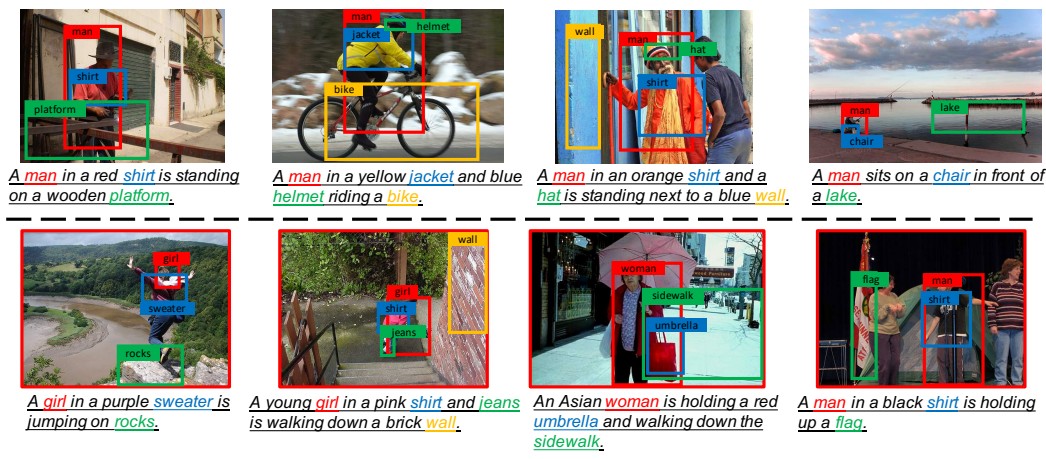

Figure 8: Correct (top) examples and examples with errors (bottom) from the proposed method.

A.4 ADDITIONAL QUALITATIVE RESULTS

In Figure 8, we show a number of correct and incorrect examples of our proposed method. We observe that while the model is able to generate grounded captions for the images, it may sometimes overlook the semantic meaning of the generated sentences, for example, *"A young girl [...] walking down a brick wall"*. Similarly, the model can overlook the spatial relationship between the objects, for instance, *"A man [...] is holding up a flag"*. While a flag is present in the scene and was able to be successfully located with the corresponding word, the man in a black shirt is spatially far from the flag.

In Figure 9, 10, 11, 12, 13, 14, 15, and 16, we illustrated the sequence of attended image region when generating each word for a complete image description. At each step, only the top-1 attended image region is shown. This is the same as how the grounding accuracy is measured. Please see the description for Figure 9 - 16 for further discussions on the qualitative results.

## A.5 Additional Implementation Details

**Region proposal features.** We use a Faster-RCNN model (Ren et al., 2015) pre-trained on Visual Genome (Krishna et al., 2017) for region proposal and feature extraction. In practice, besides the region proposal features, we also use the Conv features (*conv4*) extracted from an ImageNet pre-trained ResNet-101. Following GVD (Zhou et al., 2019), the region proposals are represented using the *grounding-aware region encoding*, which is the concatenation of i) region feature, ii) region-class similarity matrix, and iii) location embedding.

For region-class similarity matrix, we define a set of object classifiers as $\boldsymbol{W}_c$, and the region-class similarity matrix can be computed as $M_s = \text{softmax}(W_c^\top \boldsymbol{R})$, which captures the similarity between regions and object classes. We omit the ReLU and Dropout layer after the linear embedding layer for clarity. We initialize $\boldsymbol{W}_c$ using the weight from the last linear layer of an object classifiers pre-trained on the Visual Genome dataset (Krishna et al., 2017).

For location embedding, we use 4 values for the normalized spatial location. The 4-D feature is then projected to a $d_s = 300$-D location embedding for all the regions.

**Software and hardware configuration.** Our code is implemented in PyTorch. All experiments were ran on the 1080Ti, 2080Ti, and Titan Xp GPUs.

**Network architecture.** The embedding dimension for encoding the sentences is 512. We use a dropout layer with ratio 0.5 after the embedding layer. The hidden state size of the Attention and Language LSTM are 1024. The dimension of other learnable matrices are: $\boldsymbol{W}_e \in \mathbb{R}^{d_v \times 512}$, $\boldsymbol{W}_a \in \mathbb{R}^{1024 \times 512}$, $\boldsymbol{W}_{aa} \in \mathbb{R}^{512 \times 1}$, $\boldsymbol{W}_o \in \mathbb{R}^{1024 \times d_v}$, $\boldsymbol{W}_l \in \mathbb{R}^{512 \times 512}$, where the vocabulary size $d_v$ is 8639 for Flickr30k Entities and 4905 for ActivityNet-Entities.

### A.5.1 Training Details.

The hyper-parameters $\lambda_1$ and $\lambda_2$ are set to 0.5 after hyper-parameter search between 0 and 1.

**Flickr30k Entities.** Images are randomly cropped to $512 \times 512$ during training, and resized to $512 \times 512$ during inference. Before entering the proposed cyclical training regimen, the decoder was pre-trained for about 35 epochs. The total training epoch with the cyclical training regimen is around 80 epochs. The total training time takes about 1 day.

**ActivityNet-Entities.** Before entering the proposed cyclical training regimen, the decoder was pre-trained for about 50 epochs. The total training epoch with the cyclical training regimen is around 75 epochs. The total training time takes about 1 day.

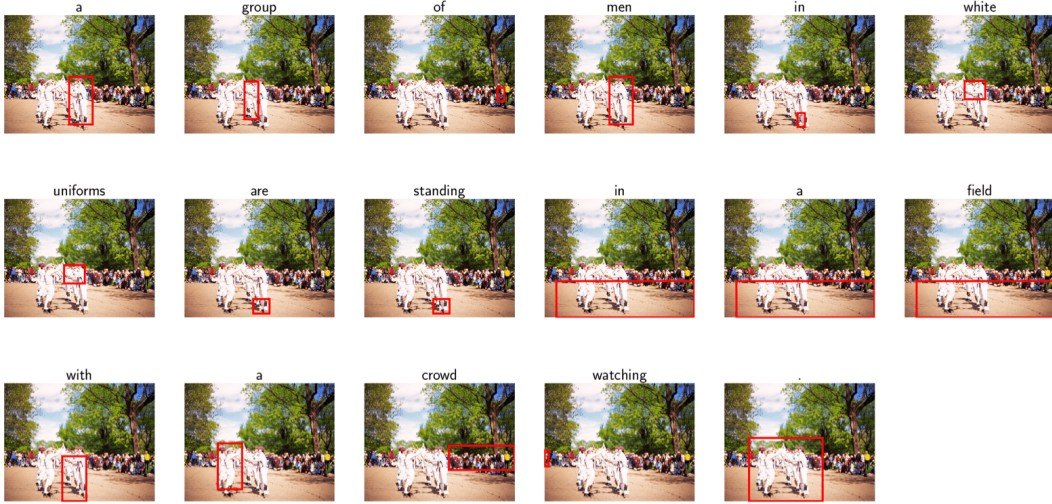

Figure 9: *A group of men in white uniforms are standing in a field with a crowd watching.* We can see that our proposed method attends to the sensible image regions for generating visually-groundable words, e.g., *man*, *uniforms*, *field*, and *crowd*. Interestingly, when generating *standing*, the model pays its attention on the image region with a foot on the ground.

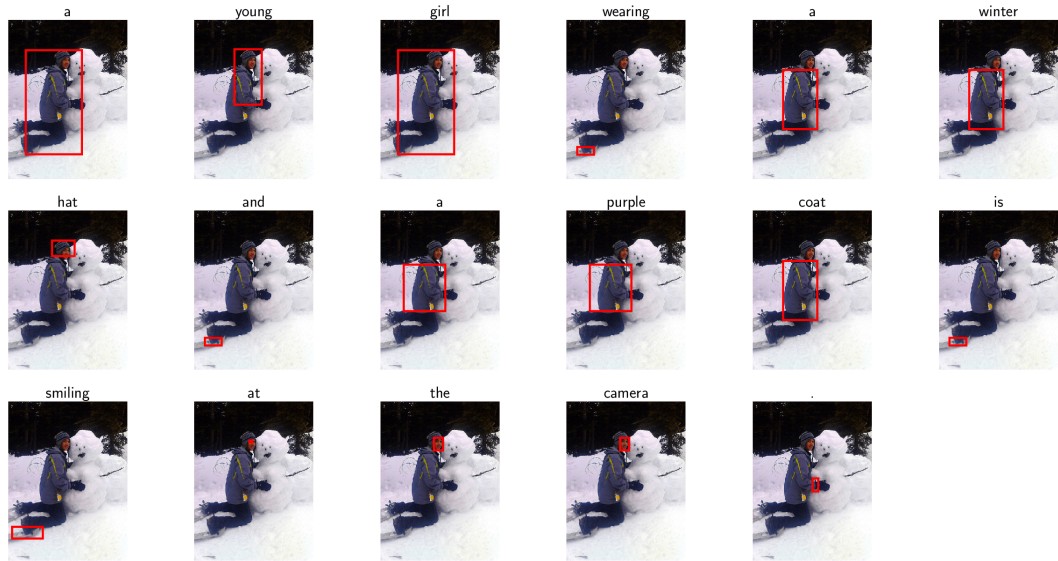

Figure 10: *A young girl wearing a winter hat and a purple coat is smiling at the camera.* The proposed method is able to select the corresponding image regions to generate *girl*, *hat*, and *coat* correctly. We have also observed that the model tends to localize the person's face when generating *camera*.

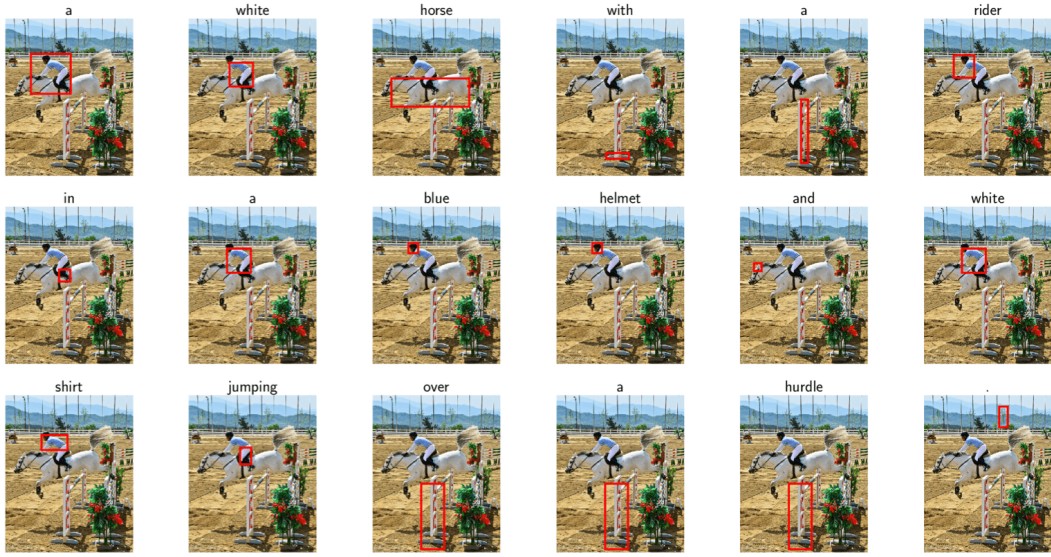

Figure 11: *A white horse with a rider in a blue helmet and white shirt jumping over a hurdle.* While the model is able to correctly locate objects such as *horse*, *rider*, *helmet*, *shirt*, and *hurdle*, it mistakenly describes the rider as wearing a blue helmet, while it's actually black, and with white shirt while it's blue.

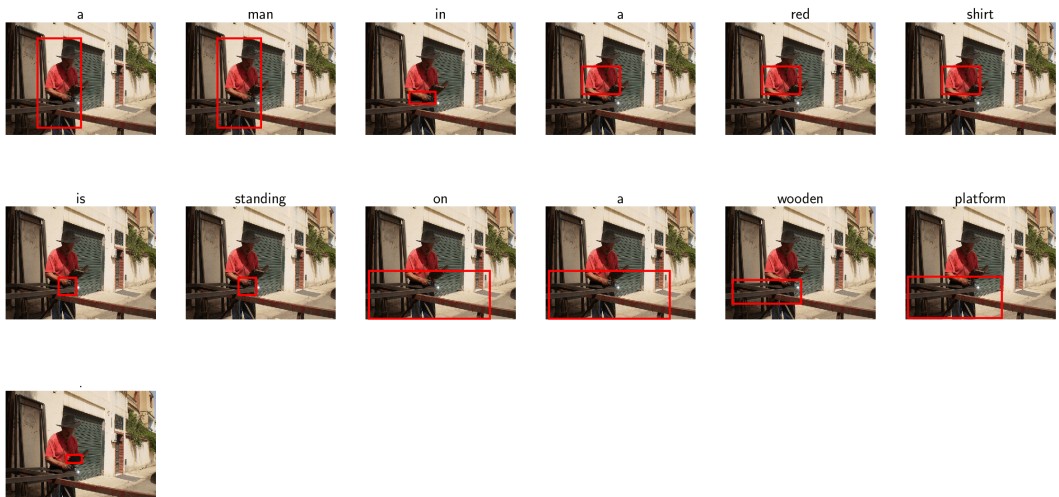

Figure 12: *A man in a red shirt is standing on a wooden platform.* Our method correctly attends on the correct regions for generating *man*, *shirt*, and *platform*.

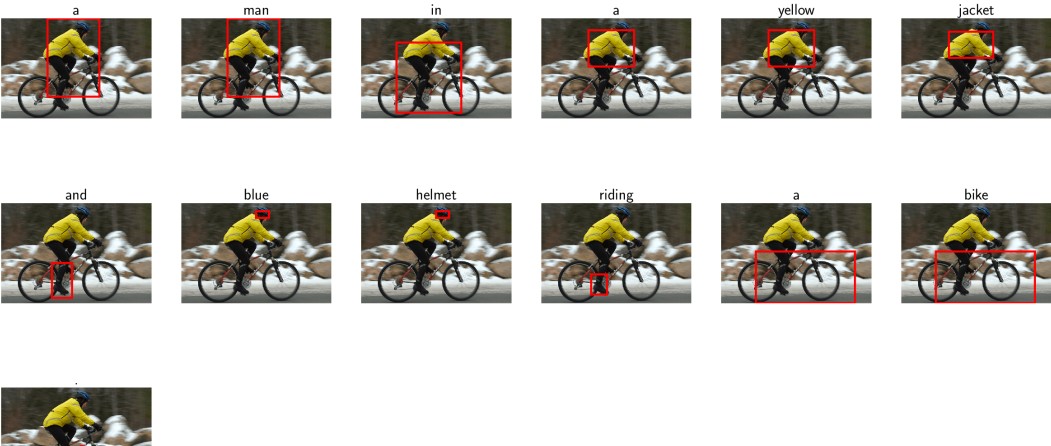

Figure 13: *A man in a yellow jacket and blue helmet riding a bike.* The proposed method correctly generates a descriptive sentence while precisely attending to the image regions for each visually-groundable words: *man*, *jacket*, *helmet*, and *bike*.

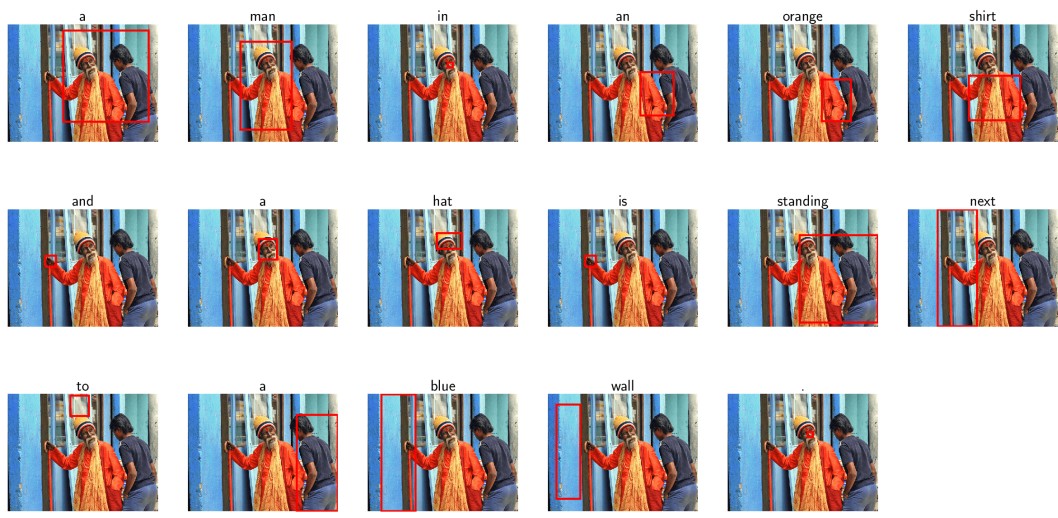

Figure 14: *A man in an orange shirt and a hat is standing next to a blue wall.* While our method is able to ground the generated sentence on the objects like: *man*, *shirt*, *hat*, and *wall* , it completely ignores the person standing next to the man in the orange cloth.

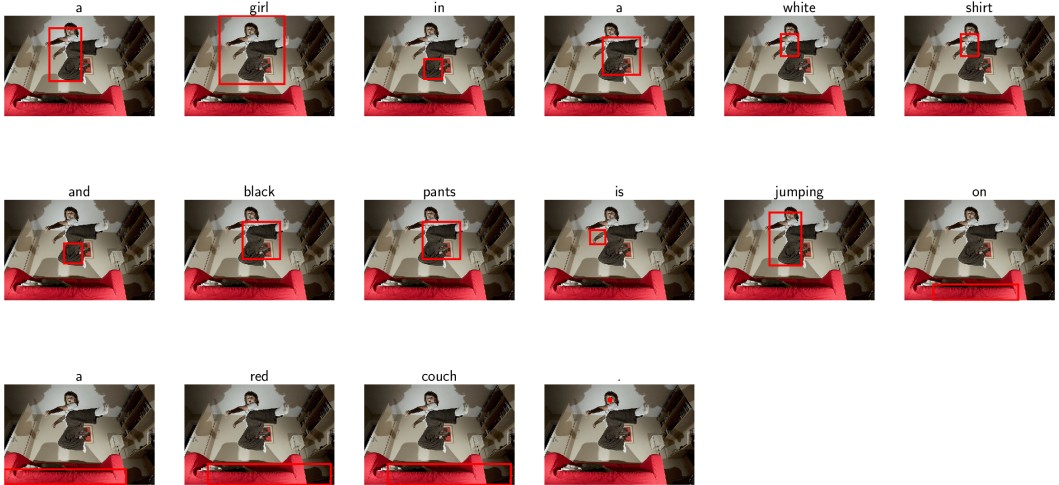

Figure 15: *A girl in a white shirt and black pants is jumping on a red couch.* Our method is able to ground the generated descriptive sentence with the correct grounding on: *girl*, *shirt*, *pants*, and *couch*.

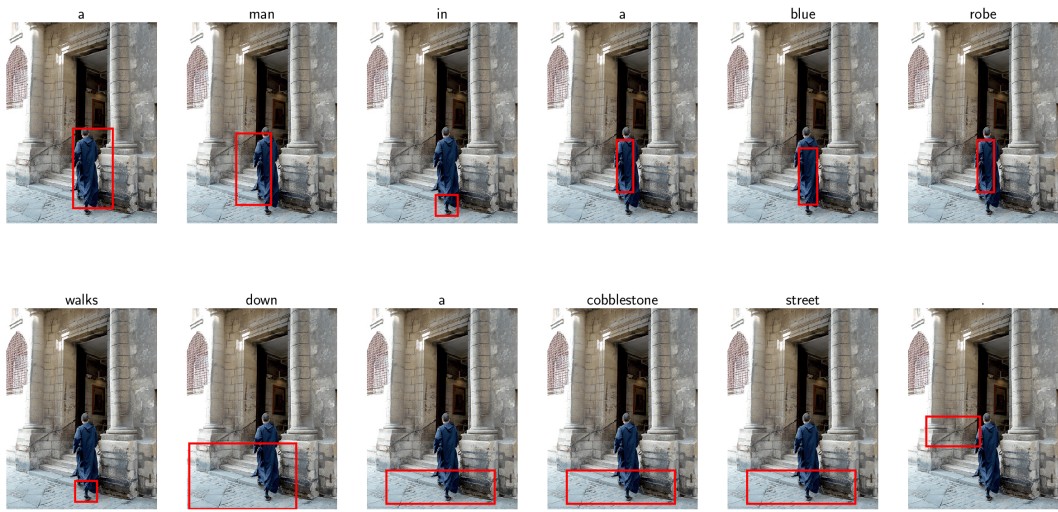

Figure 16: *A man in a blue robe walks down a cobblestone street.* Our method grounds the visually-relevant words like: *man*, *robe*, and *street*. We can also see that it is able to locate the foot on ground for *walks*.

