# OpenReview forum: "Learning to Generate Grounded Visual Captions without Localization Supervision"
_ICLR.cc/2020/Conference — Reject_

### Official Review · AnonReviewer2 · 2019-10-23
**Official Blind Review #2**

**Rating:** 3

**Review:**

# 1. Summary
The paper deals with the problem of learning grounded captions from images without joint text-location information, but instead texts (words in captions) and locations are provided independently and the model needs to figure out their link. The model is built upon GVD (Zhou et al., 2019): each word generated by the encoder is grounded to the locations provided by the region proposal module (Up-Down model (Anderson et al., 2018)), used to reconstruct the ground-truth caption.

My weak reject decision was guided by the following strengths and weaknesses of the paper.

Strengths:
* The reconstruction formulation of the problem is interesting and relevant for the task
* Proposed new metric to measure grounding performance

Weaknesses:
* Questionable motivations: it is not clear what application is grounding text to the image useful for?
* Marginal (not statistically significant?) improvement on image captioning metrics by using grounded text (proposed method) compared to not grounding (GVD)
* Limited novelty: extension of GVD, where attention is removed and object locations are used instead


# 2. Clarity and Motivation
The paper is generally well written, however there are some concerns on motivations.

One concern is related to the motivations of the paper. The authors use grounding as a proxy to improve image captioning results, which improvement is marginal wrt GVD (see Table 1). Why do we need to localize text if this has very marginal impact on the captioning metrics? It is missing the link between the potential applications where the localization of words is relevant.

The authors claim that they do not uses any grounding annotation, however the pre-trained Faster-RCNN has been trained using annotations which consist of bounding boxes + categories. Therefore, the model do (in an implicit way) rely on grounding annotations, especially because there might be an overlap between the words (classes) used to pretrain the detector and the words in the captions. The authors should assess if this ovelap/bias in the pre-trained Faster-RCNN exists or not.

Some other questions are still to be answered:
* How are the regions R parametrized? Is it the visual representation or bounding box locations?
* What happens to words that are not grounded to the image (e.g., verbs or articles)? Do you have a special way to deal with those?
* What is the intuition of multiplying word embedding and region embeddings to generate z in Eq. 6?


# 3. Novelty
The proposed method is an extension of the existing model GVD, where the attentional module is removed and its functionality is replaced by the cyclical training mode with reconstruction of the object locations. From the technical point of view this is limited novelty, but still an interesting improvement of the model; however the experiments and results do not support the claim that using such model improves image captioning result in a significant way. One way to answer to this question would have been by showing an application where the outputted locations are used for downstream tasks.


# 4. Experimentation
The experiments are carried out in a scrupulous way, by showing the comparing with GVD (with and without attention; with and without grounding supervision). The non-convincing part of them (as mentioned above already) is the fact that the improvements on these datasets might be non significant for image captioning. For example, let's consider the image captioning results in Table 1 (Flickr30k Entities): cyclical have a max improvement of 0.7 (CIDER) and min of 0 (B@4) when compared with GVD without grounding supervision. There is an obvious huge improvement on the grounding evaluation, which is obvious since GVD does not do it explicitly. The same trend is in Table 2.
These results are not convincing, combined by the fact that it is not clear in which applications one would want a very accurate grounded text.


# Minor Points
* Sec. 3.1: it is not clear that the Language LSTM is the decoder. Please explicitly say it before Eq. 1
* Caption of Table 3: which dataset is this?


**Experience Assessment:**

I have read many papers in this area.

**Review Assessment: Checking Correctness Of Derivations And Theory:**

N/A

**Review Assessment: Checking Correctness Of Experiments:**

I assessed the sensibility of the experiments.

**Review Assessment: Thoroughness In Paper Reading:**

I read the paper at least twice and used my best judgement in assessing the paper.

---

> ### Author Response · Authors · 2019-11-15
> **Clarification on goal of the work and additional results**
>
>
> We would like to thank the reviewer for the thoughtful and detailed feedback. We address the reviewer’s questions as below.
>
> --------------------------------------------------------------------------------------------------------------------------------------------------------
> 1. We focus on making captioning model more visually-grounded, not improving its captioning performance
> --------------------------------------------------------------------------------------------------------------------------------------------------------
> As mentioned in the “contributions summary” in the introduction section, we focus on improving the grounding accuracy and retaining high captioning quality. We did not claim that the proposed method would significantly improve captioning scores.
>
> Recently, visual grounding has received a great deal of attention [2] - [11]. The primary reason for this is that in many vision-and-language downstream tasks researchers have discovered that the model can learn to heavily exploit and rely on linguistic priors or dataset priors. These vision and language models suffer from poor visual grounding. They often fall back on easy-to-learn priors rather than basing model predictions on visual concepts. They do this to (seemingly) achieve good performance according to the evaluation metrics. For example, in the visual captioning task, the SoTA models can achieve CIDEr score > 1.0 on MS-COCO which is higher than the human’s 0.85 [1]. However, the quality of the generated caption is still far from the quality of those generated by humans.
>
> For this reason, we believe that there are other important aspects besides improving captioning metrics that the research community should focus on, and much existing work has been actively discussing why visual grounding is an important topic and potentially mitigate other issues in visual question answering (VQA), embedding question answering (EQA), vision-and-language navigation (VLN), visual captioning, etc [2] - [11]. Toward this direction, to the best of our knowledge, we are the first ones proposing to significantly improve grounding accuracy for captioning model without relying on the ground-truth grounding annotation and without introducing extra computation at test time. We showed that the improvements are consistent across image and video datasets. Our results in Figure 4 also indicates that our performance on rare words (which is better than the supervised method) shows that improving grounding in an unsupervised way can lead to less bias due to long-tail distribution of the grounding annotation. Finally, enforcing the model to be more visually-grounded makes the model more trustworthy and interpretable.
>
> --------------------------------------------------------------------------------------------------------------------------------------------------------
> 2. Grounding annotation and its relation to pre-trained object detector
> --------------------------------------------------------------------------------------------------------------------------------------------------------
> Grounding annotation is typically referred to as the direct links/correspondences between the words in the sentence and regions in the image/video [9][10].
>
> We argue that while the faster-RCNN is pre-trained from the object detection dataset, the pre-trained object classes do not directly translate to the words in the captions. Besides, the same image region could have different words describing it depending on the sentence context. For example, image regions on a tree could correspond to: “tree”, “forest”, “mountain”, “shrubs”, and “bushes” in sentences like: “A man is cutting a tree”, “A woman is entering a forest”, “A man is hiking in the mountain“, “A woman is trimming shrubs”, etc.
>
> If the model is biased towards the visually-groundable words (object words) in the dataset, it will, however, have poor captioning performance as we discussed in Sec 4.3, Table 3, and Table 4. Note that in all three different cases (original object detector, grounding-biased object detector, and unrealistically perfect object detector), we showed that grounding is still an important issue and our proposed method can successfully improve grounding accuracy regardless of whether the object detector is biased.

---

> > ### Author Response · Authors · 2019-11-15
> > **(Cont’d)**
> >
> >
> > --------------------------------------------------------------------------------------------------------------------------------------------------------
> > 3. How are the regions R parametrized?
> > --------------------------------------------------------------------------------------------------------------------------------------------------------
> > As we stated in Sec. 3.1 and in Appendix A.5, we follow GVD to use “grounding-aware region encoding” to encode the ROI regions, which contains the following representations: i) region visual feature, ii) region-class similarity matrix, and iii) location embedding.
> >
> > ------------------------------------------------------------------------------------------------------------------------------------------------------------------------------
> > 4. What happens to words that are not grounded to the image (e.g., verbs or articles)? Do you have a special way to deal with those?
> > ------------------------------------------------------------------------------------------------------------------------------------------------------------------------------
> > Thank you for the reviewer’s suggestion. We explored a few variants of the proposed method to deal with words that cannot be visually-grounded. We ask the reviewer to see our response to R1 for the same question.
> >
> > ------------------------------------------------------------------------------------------------------------------------------------------------------------------------------
> > 5. What is the intuition of multiplying the word embeddings and region embeddings to generate $z_{t,n}^{l}$ in Eq. 6?
> > ------------------------------------------------------------------------------------------------------------------------------------------------------------------------------
> > The localization step operates exactly as the dot-product attention selection. We simply use the learned word embedding as input for the attention selection, and the candidates for selection are the ROI regions. This is a standard way of performing widely used dot-product soft-attention [13].
> >
> > In this sense, one can also explore other variants of the attention method for the localizer. For example, we additionally ran the additive soft-attention as one variant for comparing the current method. We observed that the current dot-product attention selection performs better in terms of both captioning scores and grounding accuracy.
> >
> >
> > --------------------------------------------------------------------------------------------------------------------------------------------------------------------------
> >                                                                            Flickr30k-Entities test set (average five runs)
> > --------------------------------------------------------------------------------------------------------------------------------------------------------------------------
> >                                          B@1       B@4       M             C             S             F1_all         F1_loc             F1_all_per_sent       F1_loc_per_sent
> > --------------------------------------------------------------------------------------------------------------------------------------------------------------------------
> > Cyclical                            69.4       26.9        22.3         60.8        16.6        5.11            14.15             15.15                           35.56
> > Cyclical (localizer w/     68.9       26.0        21.9         58.4        16.0        4.31            12.02             12.88                           31.51
> > additive soft-attn)
> > --------------------------------------------------------------------------------------------------------------------------------------------------------------------------
> >
> > [1] Microsoft COCO Captions: Data Collection and Evaluation Server, 2015
> > [2] Human attention in visual question answering: Do humans and deep networks look at the same regions? EMNLP 2016
> > [3] Don't Just Assume; Look and Answer: Overcoming Priors for Visual Question Answering, CVPR 2018
> > [4] Overcoming Language Priors in Visual Question Answering with Adversarial Regularization, NeurIPS 2018
> > [5] Quantifying and alleviating the language prior problem in visual question answering, SIGIR 2019
> > [6] Blindfold Baselines for Embodied QA, NeurIPS 2018 ViGilL Workshop
> > [7] Attention correctness in neural image captioning, AAAI 2017.
> > [8] Object hallucination in image captioning, EMNLP 2018.
> > [9] Grounded Video Description, CVPR 2019
> > [10] Taking a HINT: Leveraging Explanations to Make Vision and Language Models More Grounded, ICCV 2019
> > [11] Self-Monitoring Navigation Agent via Auxiliary Progress Estimation, ICLR 2019
> > [12] Are You Looking? Grounding to Multiple Modalities in Vision-and-Language Navigation, ACL 2019.
> > [13] Attention Is All You Need, NeurIPS 2017.

---

### Official Review · AnonReviewer3 · 2019-10-24
**Official Blind Review #3**

**Rating:** 6

**Review:**

This paper addresses captioning generation for images and videos, by proposing a novel cyclical training regimen consisting of three steps: decoding, localization, and reconstruction. The experimental results show that the performance on image captioning and video captioning are improved without grounding supervision.

I lean to accept this paper. The motivation using cyclic feedback itself is not so novel for language generation, but focusing on grounding without localization supervision for visual captioning is interesting. The experimental results show that the proposed method can boost performance both qualitatively and qualitatively. I have several comments and questions below.
- Why do the authors introduce GVD without self-attention as a baseline? Table 1 and 2 show that removing self-attention degrades the performance. If the combination of self-attention in GVD and cyclical training proposed in this paper is complementary to each other, it does help to improve the overall accuracy.
- While the authors develop a cyclical training pipeline, including decoding, localization, and reconstruction, Figure 1 does not show which part corresponds to the decoding phase. The authors should clarify it to make the paper easier to be understood.
- Equation (5) seems to be strange. $\theta^*$, a sum of two parameters for each arg max operator, doesn't guarantee that each term in the right side of Eq. (5) keeps its max. This equation seems to be a conceptual one, and the actual training would be performed according to Eq. (7). Therefore, the experimental results might not be influenced by the error in Eq. (5).
- $\hat{r}^l_t=\beta_t^\top R$ between Eq. (6) and Eq. (7) means that $\hat{r}^l_t$ is a row vector while $r_n$ seems to be a column vector. Since $R = [r_1, r_2, ..., r_N]$ for $N$ regions, $\hat{r}^l_t= R \beta_t$ seems to be appropriate. The authors should correct it. I have a similar comment for $\hat{r}_t = \alpha_t^\top R$ in Eq. (2).
- In the caption of Table 5, the number equal to or smaller than ten should be spelled out; "5 runs" should be "five runs." There are similar errors, such as "5 GT captions" and "1 GT caption" in Sec. 4.
- The format of items in References is not consistent.
- According to Sec. A.4.1, $\lambda_1$ and $\lambda_2$ are tuned between 0 and 1. How are the experimental results sensitive to these hyperparameters? Additional experiments using different $\lambda_1$ and $\lambda_2$ would be helpful.

**Experience Assessment:**

I have published in this field for several years.

**Review Assessment: Checking Correctness Of Derivations And Theory:**

I carefully checked the derivations and theory.

**Review Assessment: Checking Correctness Of Experiments:**

I carefully checked the experiments.

**Review Assessment: Thoroughness In Paper Reading:**

I read the paper thoroughly.

---

> ### Author Response · Authors · 2019-11-15
> **Baseline clarification and experimental results of various loss weightings**
>
>
> We would like to thank the reviewer for the thoughtful, constructive feedback, and especially paying extra attention to the details. With the reviewer’s suggestions, we have clarified the caption of Figure 1, corrected the equations, and made the format of the references consistent. Please see the revised pdf version.
>
> We address the other questions as below.
> --------------------------------------------------------------------------------------------------------------------------------------------------------
> 1. Why do the authors introduce GVD without self-attention as a baseline?
> --------------------------------------------------------------------------------------------------------------------------------------------------------
> From Table 2 in the GVD paper (also copied below), removing the self-attention does not seem to degrade the performance. In fact, both captioning and grounding accuracy seem to be better without self-attention. What is more,  grounding accuracy is also significantly better without self-attention. We thus decided to use the model without self-attention as our baseline model. When comparing our baseline results (averaged across five runs) with GVD (only single run), they are comparable in terms of both captioning and grounding performances. To make it clear that our baseline is still comparable with GVD, we also report the standard deviation across five runs in the tables below.
>
> --------------------------------------------------------------------------------------------------------------------------------------------------------
>                                                                            ActivityNet Entities val set
> --------------------------------------------------------------------------------------------------------------------------------------------------------
>                                                              B@1          B@4           M                C                 S                F1_all          F1_loc
> --------------------------------------------------------------------------------------------------------------------------------------------------------
> GVD (single run)                               23.0           2.27           10.7            44.6            13.8           0.28             1.13
> GVD (single run) (w/o SelfAttn)     23.2           2.28           10.9            45.6            15.0           3.70             12.7
> Baseline (average five runs)           23.2±0.5    2.22±0.2    10.8±0.3    45.9±1.5    15.1±0.2    3.75±0.16   12.00±0.76
> --------------------------------------------------------------------------------------------------------------------------------------------------------
>
> --------------------------------------------------------------------------------------------------------------------------------------------------------
>                                                                            Flickr30k-Entities test set
> --------------------------------------------------------------------------------------------------------------------------------------------------------
>                                                              B@1          B@4           M                C                 S                F1_all          F1_loc
> --------------------------------------------------------------------------------------------------------------------------------------------------------
> GVD (single run)                               69.2           27.3           22.5            60.1            16.1           3.97             11.6
> Baseline (average five runs)          69.1±0.6    26.0±0.6    22.1±0.3    59.6±0.6     16.3±0.2   4.08±0.40    11.83±1.27
> --------------------------------------------------------------------------------------------------------------------------------------------------------

---

> > ### Author Response · Authors · 2019-11-15
> > **(Cont’d)**
> >
> >
> > --------------------------------------------------------------------------------------------------------------------------------------------------------
> > 2. Weighting between decoding and reconstruction losses
> > --------------------------------------------------------------------------------------------------------------------------------------------------------
> > The weighting between the two losses was chosen with a preliminary grid search early when developing the proposed method. With the reviewer’s suggestion, we re-ran the search for the loss weighting and report the numbers below. We can see that when comparing to the baseline, all different loss weightings consistently improved the grounding accuracy.
> >
> > --------------------------------------------------------------------------------------------------------------------------------------------------------------------------
> >                                                                            Flickr30k-Entities test set (average five runs)
> > --------------------------------------------------------------------------------------------------------------------------------------------------------------------------
> > [$\lambda_1$, $\lambda_2$]            B@1            B@4            M             C             S           F1_all           F1_loc            F1_all_per_sent            F1_loc_per_sent
> > --------------------------------------------------------------------------------------------------------------------------------------------------------------------------
> > Baseline            69.1            26.0            22.1         59.6        16.3        4.08            11.83             13.20                              31.83
> > --------------------------------------------------------------------------------------------------------------------------------------------------------------------------
> > [0.8, 0.2]            69.6            26.9            22.4         60.9        16.7        4.95            14.03             14.67                              34.72
> > [0.6, 0.4]            69.9            27.4            22.3         61.4        16.6        4.98            13.53             15.03                              35.54
> > [0.5, 0.5]            69.4            26.9            22.3         60.8        16.6        5.11            14.15             15.15                              35.56
> > [0.4, 0.6]            69.6            27.3            22.6         61.9        16.8        4.92            13.53             15.97                              37.57
> > [0.2, 0.8]            69.2            27.0            22.4         61.1        16.5        4.88            13.62             16.20                              39.05
> > --------------------------------------------------------------------------------------------------------------------------------------------------------------------------

---

### Official Review · AnonReviewer1 · 2019-10-25
**Official Blind Review #1**

**Rating:** 6

**Review:**

The paper proposes an architecture that grounds words from a captioning model, but without requiring explicit per-word grounding training data. Instead, they show that it is sufficient to use cycle consistency, verifying that by predicting word->grounding->word the two words are the same.

General:

Cycle consistency has been shown to be very useful in replacing explicit paired data, for eample in image-to-image translation (CycleGAN, or the more recent FUNIT). This paper takes it to the domain of vision and language. While the novelty is not very large it seems like a solid step in an interesting direction.  Evaluated on both image and video captioning with substantial localization improvement in the specific relevant eval settings.

Specific comments:

-- In Table 1, the part of "Caption Evaluation" the proposed method is in bold, but it seems that "Up-Down" method out-performs the proposed method in B@1 and B@4.

-- Are words that are not nouns/verbs (the/a/are/with/etc) handled differently? It doesn't really make sense to localize them just like object words.

-- The localization model is linear? What would be the effect of richer models on localization accuracy?

-- Qualitative analysis: It would have been useful to add evaluations by human-raters to measure the perceptual quality of the localization.

-- Error analysis? examples and analysis  of failure cases?



**Experience Assessment:**

I have published one or two papers in this area.

**Review Assessment: Checking Correctness Of Derivations And Theory:**

I assessed the sensibility of the derivations and theory.

**Review Assessment: Checking Correctness Of Experiments:**

I assessed the sensibility of the experiments.

**Review Assessment: Thoroughness In Paper Reading:**

I read the paper at least twice and used my best judgement in assessing the paper.

---

> ### Author Response · Authors · 2019-11-15
> **Additional experimental results and human evaluation on grounding**
>
>
> We would like to thank the reviewer for the thoughtful and constructive feedback. We address the reviewer’s questions as below.
>
> --------------------------------------------------------------------------------------------------------------------------------------------------------
> 1. Are words that are not nouns/verbs handled differently?
> --------------------------------------------------------------------------------------------------------------------------------------------------------
> In the currently proposed method, all the words are handled the same regardless of whether they are nouns or verbs. We thank the reviewers (R1 and R2) for the comment, and we explored a few variants to handle nouns and verbs differently from other words (such as articles) as suggested. The experimental results, however, were mixed, sometimes better and sometimes worse. Mainly, we explored with two variants:
>         (1) The reconstruction loss is only computed when the target word is either nouns or verbs.
>         (2) The localized region representation will be invalid (set to zero) if the target word is neither nouns nor verbs.
>
> For the first variant (1), we observed that the captioning performance stays the same while grounding accuracy has a small improvement. On the other hand, for the second variant (2), we can see that all captioning scores are improved over baseline with CIDEr improved 2.4 (averaged across five runs). We can also see that grounding accuracy on per sentence basis further improved as well. We then conducted further experiments on both ActivityNet-Entities and Flickr30k Entities with unrealistically perfect object detector (see tables below), the improvements however are not consistent. In summary:
>         1) On the Flickr30k Entities test set, we observed that CIDEr is better and grounding per sentence is also better
>         2) On the ActivityNet-Entities val set, the captioning performances are about the same but grounding accuracy became worse.
>         3) On the Flickr30k Entities test set with unrealistically perfect object detector, captioning performances are slightly worse but grounding accuracies are improved.
>
> We have included the additional experimental results/discussions of Flickr30k Entities and ActivityNet-Entities in Tables 6, 7, and 8 in the Appendix in the revision.
>
> --------------------------------------------------------------------------------------------------------------------------------------------------------------------------
>                                                                           Flickr30k-Entities test set (average five runs)
> --------------------------------------------------------------------------------------------------------------------------------------------------------------------------
>                                          B@1       B@4       M             C             S             F1_all         F1_loc             F1_all_per_sent       F1_loc_per_sent
> --------------------------------------------------------------------------------------------------------------------------------------------------------------------------
> Baseline                          69.1       26.0        22.1         59.6        16.3        4.08            11.83             13.20                           31.83
> Cyclical                            69.4       26.9        22.3         60.8        16.6        5.11            14.15             15.15                           35.56
> Cyclical (1)                      69.7       27.0        22.2         60.1        16.5        5.14            14.32             15.36                           36.33
> Cyclical (2)                      69.9       27.5        22.4         62.0        16.6        5.13            13.99             16.30                           38.45
> --------------------------------------------------------------------------------------------------------------------------------------------------------------------------

---

> > ### Author Response · Authors · 2019-11-15
> > **(Cont’d)**
> >
> >
> > --------------------------------------------------------------------------------------------------------------------------------------------------------
> > 4. Error analysis
> > --------------------------------------------------------------------------------------------------------------------------------------------------------
> > We show several examples of failure cases in the bottom row of Figure 8 and the corresponding discussions are provided in Sec. A.4. In particular, we believe that the model sometimes overlooks the semantic meaning of the generated captions though it was able to ground the corresponding objects in the scene. For example, it says “a young girl is walking down a brick wall”, where “wall” is correctly localized, but “walking down a wall” is unlikely to happen.
> >
> > --------------------------------------------------------------------------------------------------------------------------------------------------------
> > 4. Suggestion on bold number in Table 1 and 2
> > --------------------------------------------------------------------------------------------------------------------------------------------------------
> > To make a fair comparison, we only bolded the numbers that are averaged across multiple runs. This is because our baseline is already a direct replica with the SoTA of GVD (without self-attention), and given the variances of the experimental results for visual captioning tasks, we believe it is best to report and compare experimental results across multiple runs. We have also corrected table caption to make it clear. However, if the reviewer feels strongly about this, we are happy to change it according to the reviewers suggestion in the final revision.

---

> > ### Author Response · Authors · 2019-11-15
> > **(Cont’d)**
> >
> >
> > --------------------------------------------------------------------------------------------------------------------------------------------------------
> > 2. Can non-linear localizer improve grounding accuracy?
> > --------------------------------------------------------------------------------------------------------------------------------------------------------
> > Our previous experiments showed that using a multilayer perceptron (MLP) for the localizer instead reduced both captioning and grounding accuracy. Monitoring the losses and various metrics during training, it is clear that when using MLP, the model tends to overfit. We report the experimental results in the table below:
> >
> > --------------------------------------------------------------------------------------------------------------------------------------------------------------------------
> >                                                                           Flickr30k Entities test set (average five runs)
> > --------------------------------------------------------------------------------------------------------------------------------------------------------------------------
> >                                              B@1       B@4       M             C             S             F1_all         F1_loc          F1_all_per_sent       F1_loc_per_sent
> > --------------------------------------------------------------------------------------------------------------------------------------------------------------------------
> > Cyclical                                69.4       26.9        22.3         60.8        16.6        5.11          14.15             15.15                         35.56
> > Cyclical (MLP Localizer)    69.2       26.4        22.0         58.7        16.2        4.40          12.77             13.97                         33.40
> > --------------------------------------------------------------------------------------------------------------------------------------------------------------------------
> >
> > --------------------------------------------------------------------------------------------------------------------------------------------------------
> > 3. Human evaluation on the perceptual quality of the localization/grounding
> > --------------------------------------------------------------------------------------------------------------------------------------------------------
> > As suggested, we conduct a human evaluation on the perceptual quality of the grounding. We asked 10 human subjects to pick the best among two grounded regions (by baseline and Cyclical) for each word. The subjects have three options to choose from: 1) grounded region A is better, 2) grounded region B is better, and 3) they are about the same. Authors or other colleagues familiar with the proposed method were excluded from the study. Each of the human subjects were given 25 images, each with a varying number of groundable words. Each image was presented to two different human subjects in order to be able to measure inter-rater agreement. To avoid being biased towards the “object words” defined in the dataset for automatic grounding evaluation, for the study we define a word to be groundable if it is either a noun or verb. The order of approaches was randomized for each sentence. Please see Figure 7 in the revision for our demonstration of human evaluation study.
> >
> > Our experiment on the Flickr30k Entities val set showed that:
> >         28.1% of words are more grounded by Cyclical
> >         24.8% of words are more grounded by baseline
> >         47.1% of words are similarly grounded
> >
> > Regarding the inter-rater agreement between each pair of human subjects:
> >         72.7% of ratings are the same
> >         4.9% of ratings are the opposite
> >         22.4% of ratings could be ambiguous (e.g., one chose A is better, the other chose they are about the same)
> >
> > We would also like to make a note that the grounded words judged to be similar largely consisted of very easy or impossible cases. For example, words like "mountain", "water", “street", etc, are typically rated to be "about the same" since they usually have many possible boxes and is very easy for both models to ground the words correctly. On the other hand, for visually ungroundable cases: E.g. "stand" appears a lot and the subject would choose "about the same" since the image doesn't cover the fact that the person's feet are on the ground.
> >
> > We see that the human study results follow the grounding results presented in the paper and show an improvement in grounding accuracy for the proposed method over a strong baseline. The improvement is achieved without grounding annotations or extra computation at test time. We have included the results and a demo figure of human evaluation in the revision (see Sec. A.3 and Figure 7).

---

> > ### Author Response · Authors · 2019-11-15
> > **(Cont’d)**
> >
> >
> > --------------------------------------------------------------------------------------------------------------------------------------------------------------------------
> >                                                                           ActivityNet Entities val set (average five runs)
> > --------------------------------------------------------------------------------------------------------------------------------------------------------------------------
> >                                          B@1       B@4       M             C             S             F1_all         F1_loc             F1_all_per_sent       F1_loc_per_sent
> > --------------------------------------------------------------------------------------------------------------------------------------------------------------------------
> > Baseline                          23.2       2.22        10.8         45.9        15.1        3.75            12.00             9.41                           31.68
> > Cyclical                            23.7       2.45        11.1         46.4        14.8        4.68            15.84             12.60                           44.04
> > Cyclical (2)                      23.9       2.58        11.2         46.6        14.8        4.48            15.01             11.53                           40.30
> > --------------------------------------------------------------------------------------------------------------------------------------------------------------------------
> >
> > --------------------------------------------------------------------------------------------------------------------------------------------------------------------------
> >                                                                           Flickr30k Entities test set (average three runs)
> > --------------------------------------------------------------------------------------------------------------------------------------------------------------------------
> >                                          B@1       B@4       M             C             S             F1_all         F1_loc             F1_all_per_sent       F1_loc_per_sent
> > --------------------------------------------------------------------------------------------------------------------------------------------------------------------------
> > Unrealistically perfect object detector
> > --------------------------------------------------------------------------------------------------------------------------------------------------------------------------
> > Baseline                          75.1       32.1        25.2         76.3        22.0        20.82          48.74             43.21                           77.81
> > Cyclical                            76.7       32.8        25.8         80.2        22.7        25.27          54.54             46.98                           81.56
> > Cyclical (2)                      75.8       32.2        25.6         79.0        22.4        25.65          55.81             48.99                           85.99
> > --------------------------------------------------------------------------------------------------------------------------------------------------------------------------

---

### Decision · Program_Chairs · 2019-12-19

**Decision:**

Reject

**Comment:**

This paper proposes a cyclical training scheme for grounded visual captioning, where a localization model is trained to identify the regions in the image referred to by caption words, and a reconstruction step is added conditioned on this information. This extends prior work which required grounding supervision.

While the proposed approach is sensible and grounding of generated captions is an important requirement, some reviewers (me included) pointed out concerns about the relevance of this paper's contributions. I found the authors’ explanation that the objective is not to improve the captioning accuracy but to refine its grounding performance without any localization supervision a bit unconvincing -- I would expect that better grounding would be reflected in overall better captioning performance, which seems to have happened with the supervised model of Zhou et al. (2019). In fact, even the localization gains seem rather small: “The attention accuracy for localizer is 20.4% and is higher than the 19.3% from the decoder at the end of training.” Overall, the proposed model is an incremental change on the training of an image captioning system, by adding a localizer component, which is not used at test time. The authors' claim that “The network is implicitly regularized to update its attention mechanism to match with the localized image regions” is also unclear to me -- there is nothing in the loss function that penalizes the difference between these two attentions, as the gradient doesn’t backprop from one component to another. Sharing the LSTM and Language LSTM doesn’t imply this, as the localizer is just providing guidance to the decoder, but there is no reason this will help the attention of the original model.

Other natural questions left unanswered by this paper are:
- What happens if we use the localizer also in test time (calling the decoder twice)? Will the captions improve? This experiment would be needed to assess the potential of this method to help image captioning.
- Can we keep refining this iteratively?
- Can we add a loss term on the disagreement of the two attentions to actually achieve the said regularisation effect?

Finally, the paper [1] (cited by the authors) seems to employ a similar strategy (encoder-decoder with reconstructor) with shown benefits in video captioning.

[1] Bairui Wang, Lin Ma, Wei Zhang, and Wei Liu. Reconstruction network for video captioning. In Proceedings of the IEEE Conference on Computer Vision and Pattern Recognition (CVPR), pp. 7622–7631, 2018.

I suggest addressing some of these concerns in a revised version of the paper.